

# Exploring the technological dimension of Autonomous sensory meridian response-induced physiological responses

Sahar Seifzadeh[1] and Bozena Kostek[2]

[1] Doctoral School of Gdańsk University of Technology, Faculty of Electronics, Telecommunications and Informatics, Multimedia Systems Department, Gdańsk University of Technology, Gdansk, N/A, Poland
[2] Faculty of Electronics, Telecommunications and Informatics, Audio Acoustics Laboratory, Gdańsk University of Technology, Gdansk, N/A, Poland

Corresponding author
Bozena Kostek,
bokostek@audioakustyka.org

## ABSTRACT

**Background**. In recent years, the scientific community has been captivated by the intriguing Autonomous sensory meridian response (ASMR), a unique phenomenon characterized by tingling sensations originating from the scalp and propagating down the spine. While anecdotal evidence suggests the therapeutic potential of ASMR, the field has witnessed a surge of scientific interest, particularly through the use of neuroimaging techniques including functional magnetic resonance imaging (fMRI) as well as electroencephalography (EEG) and physiological measures such as eye tracking (Pupil Diameter), heart rate (HR), heartbeat-evoked potential (HEP), blood pressure (BP), pulse rates (PR), finger photoplethysmography (PPG), and skin conductance (SC). This article is intended to provide a comprehensive overview of technology's contributions to the scientific elucidation of ASMR mechanisms.

**Methodology**. A meticulous literature review was undertaken to identify studies that have examined ASMR using EEG and physiological measurements. The comprehensive search was conducted across databases such as PUBMED, SCOPUS, and IEEE, using a range of relevant keywords such as 'ASMR', 'Autonomous sensory meridian response', 'EEG', 'fMRI', 'electroencephalography', 'physiological measures', 'heart rate', 'skin conductance', and 'eye tracking'. This rigorous process yielded a substantial number of 63 PUBMED and 166 SCOPUS-related articles, ensuring the inclusion of a wide range of high-quality research in this review.

**Results**. The review uncovered a body of research utilizing EEG and physiological measures to explore ASMR's effects. EEG studies have revealed distinct patterns of brain activity associated with ASMR experiences, particularly in regions implicated in emotional processing and sensory integration. In physiological measurements, a decrease in HR and an increase in SC and pupil diameter indicate relaxation and increased attention during ASMR-triggered stimuli.

**Conclusions**. The findings of this review underscore the significance of EEG and physiological measures in unraveling the psychological and physiological effects of ASMR. ASMR experiences have been associated with unique neural signatures, while physiological measures provide valuable insights into the autonomic responses elicited by ASMR stimuli. This review not only highlights the interdisciplinary nature of ASMR research but also emphasizes the need for further investigation to elucidate the mechanisms underlying ASMR and explore its potential therapeutic applications, thereby paving the way for the development of novel therapeutic interventions.

## INTRODUCTION

In 2007, the sensory phenomenon known as Autonomous sensory meridian response (ASMR) was first acknowledged during discussions on an online health forum, marking its initial recognition (*Trenholm-Jensen et al., 2022*). ASMR, identified as a sensory-emotional phenomenon (*Del Campo & Kehle, 2016*), is purported to induce calming and gratifying sensations in the scalp, neck, and occasionally in the back and spine in response to stimuli such as auditory cues (*e.g.*, whispers), visual stimuli (*e.g.*, observing hair brushing), or tactile (physical touch or sensation of being touched). While all these stimuli are processed through sensory pathways, auditory and visual stimuli do not require direct contact with the body, whereas tactile stimuli are experienced through direct (or imagined) physical contact. Moreover, sound and visuals may activate brain regions involved in emotional processing and attention, such as the prefrontal cortex and temporal lobes. In contrast, tactile stimuli directly engage the somatosensory cortex, which is responsible for processing physical touch sensations, potentially leading to a different type and intensity of ASMR experience. That is why, in our review, we focused on auditory and visual stimuli delivered by video. It should, however, be noted that ASMR extends beyond videos; it is also part of daily experiences. ASMR sensations can be triggered in real life through whispered voices, soft tapping noises, calming visual patterns, or gentle touches. These stimuli often provide relaxation and a soothing, tingling feeling. Hence, ASMR videos aim to recreate and simulate real-world ASMR experiences that occur naturally through various sensory stimuli such as sound, visuals, and touch.

As of December 2023, ASMR has emerged as an unparalleled phenomenon on YouTube dominating the platform as the most searched term. The scale of its popularity is enormous, with an astonishing monthly search volume that has surged to 67.4 million queries per month. Such an interest demonstrates the immense appeal that ASMR content holds for viewers worldwide. What makes this trend particularly notable is its stability and longevity. ASMR first began making waves on YouTube several years ago, and it has shown remarkable staying power. In fact, it has maintained its position on the YouTube search rankings, as evidenced by the second most searched term on the platform in 2021, right before it claimed the top spot in 2023 (*Similarweb, 2023*).

Certainly, it is decisively more important that the significant number of recent research articles clearly demonstrates the enduring and widespread interest in ASMR content. Specifically, a search on PubMed yielded 63 ASMR articles, while Scopus returned 166 articles, and IEEE published 14 ASMR-related articles. These findings underscore the growing scholarly attention devoted to ASMR. When further looking into the details, these articles belong to psychology, medicine, neuroscience, computer sciences, and social sciences. Also, ASMR is the most frequent keyword within these articles (keywords are

shown in Table 1). Therefore, one motivation for undertaking a review on ASMR is the considerable and recent surge in scientific attention and importance of this subject.

To delve into the impact of ASMR, this phenomenon is studied through psychological, physiological, cognitive, and neurological lenses. From a psychological perspective, investigations have begun to examine the variations in ASMR sensitivity and preferences among individuals.

There are countless ASMR triggers in ASMR videos on YouTube, such as whispering, soft-spoken, eating/chewing sounds, hair play, binaural role-plays, tapping, crinkling, page-turning, tapping fingernails, scratching, brushing, personal attention, hand movements, and mic brushing. However, the most effective, as reported, are whispering (75%), personal attention (69%), and crisp sounds (64%), respectively. The findings of this particular study indicate that 98% of participants used ASMR to alleviate anxiety and experience peace, 82% to increase sleep, and 70% to reduce stress (*Barratt & Davis, 2015*). Factors such as personality traits, prior experiences, and cultural background may influence an individual's susceptibility to ASMR triggers and the intensity of their ASMR experiences. For this purpose, two instruments were highlighted: the ASMR-15 (*Roberts, Beath & Boag, 2019*) and the ASMR-Experience (*Swart et al., 2022*). The ASMR-15 is a traditional 15-item questionnaire categorizing ASMR experiences, while the AEQ is an online questionnaire with 14 questions related to ASMR videos, including a body map for detailed bodily experiences. Moreover, experiencing ASMR is associated with a positive mood, marked by relaxation. A study by *Poerio, Mank & Hostler (2022)* involved 1,002 participants, with 813 experiencing ASMR. Those sensitive to ASMR reported increased excitement, calmness, and reduced stress and sadness after watching ASMR videos compared to non-sensitive participants. Differences in response to ASMR were further pursued, and the results indicated an association between how individuals respond to their environment, their interoception abilities, and the way they evaluate emotions (*Poerio, Mank & Hostler, 2022*). The brain's map for pleasant touch has been found to cross-activate in ASMR. This cross-activation is believed to contribute to subjective wellbeing and heightened parasympathetic activity. Conversely, in conditions such as misophonia, this same cross-activation leads to decreased emotional wellbeing and heightened sympathetic activity. This hypothesis not only sheds light on the relationship between sensory experiences and wellbeing. It also offers insights into why individuals may experience diminished wellbeing in modern urban environments, which often feature unpleasant sounds (*McGeoch & Rouw, 2020*).

Employing the sensory suggestibility scale, individuals who experience ASMR are more likely to have imaginary sensory experiences (*Keizer et al., 2020*). Meanwhile, some compare ASMR to frisson, known as aesthetic or musical chills (*Craig (Bud), 2009*; *Craig (Bud), 2005*), which involves strong emotional reactions to impactful stimuli like music. Unlike frisson, ASMR typically produces calming and tingling sensations from audio-visual triggers. Regarding the perception of time (*Seifzadeh & Nazari, 2019*) compared time perception between ASMR and music. The ASMR video featured five clips with popular YouTube triggers, while the music video included five tracks from different memorable genres, lasting twelve minutes. Participants in each group verbally estimated the video's duration, with the ASMR group reporting an average perceived time of approximately

**Table 1 ASMR keywords analysis.** The common feature among 63 ASMR articles returned by PubMed and 166 Scopus published articles is that they are very recent ones. When further looking into the details, these articles belong to psychology, medicine, neuroscience, computer sciences, and social sciences. Also, ASMR is the most frequent keyword within these articles.

| Keywords for ASMR and EEG | Keywords for ASMR and fMRI | Keywords for ASMR and HR | Keywords for ASMR and SC | ASMR and Eye Tracking |
|---|---|---|---|---|
| ASMR EEG studies | ASMR and fMRI | ASMR and cardiovascular response | ASMR and electrodermal activity | ASMR gaze behavior |
| EEG analysis of ASMR | Brain imaging | Heart rate variability during ASMR | Skin conductance during ASMR | Eye tracking during ASMR |
| Brainwaves during ASMR | fMRI ASMR analysis | Autonomic nervous system and ASMR | Autonomic arousal and ASMR | Visual attention in ASMR |
| EEG correlates of ASMR | during ASMR | Cardiovascular changes in response to ASMR | Emotional responses and skin conductance in ASMR | Oculomotor responses to ASMR triggers |
| Neural responses to ASMR triggers | Brain scan and ASMR | | ASMR triggers and skin conductance levels | Eye movement patterns in ASMR experiences |
| | fMRI ASMR triggers | | | |
| | ASMR neural pathways fMRI | | | |
| | Brain activation during ASMR fMRI | | | |
| | ASMR fMRI brain connectivity | | | |
| Keywords For EEG | Keywords For fMRI | Keywords for HR | Keywords for SC | Keywords for Eye Tracking |
| Electroencephalography | Functional Magnetic Resonance Imaging | Heart rate | Skin conductance | Eye tracking |
| Brainwaves | Brain imaging techniques | Heart rate variability (HRV) | Electrodermal activity (EDA) | Oculomotor behavior |
| Neurofeedback | Neuroimaging methods | Pulse rate | Galvanic skin response (GSR) | Gaze analysis |
| EEG headset | Brain activity measurement | Cardiac activity | Sudomotor activity | Pupil dilation |
| Alpha waves | Functional brain mapping | Autonomic nervous system | Skin conductance level | Saccades |
| Beta waves | Functional MRI scan | Sympathetic nervous system | of Emotional arousal | Fixations |
| Delta waves | Functional | Parasympathetic nervous system | Autonomic nervous system | |
| Theta waves | | Cardiovascular responses | Physiological responses | |
| Neuroimaging | | Physiological arousal | | |
| EEG data | | Cardiophysiology | | |

eleven minutes and the music group around twelve minutes. An independent $t$-test showed no significant difference between the two groups' time estimations.

Our article is driven by two primary motivations and an additional recent ASMR influence seen in research. First, we aim to examine existing ASMR-related research by utilizing the technological resources available in our laboratory, including a comprehensive review of relevant literature with the tools and capabilities we and other labs possess. Second, we strive to present our review in a way that is accessible to those new to ASMR research, particularly those with backgrounds in engineering, to facilitate interdisciplinary dialogue and potentially unveil insights into this intriguing phenomenon. In addition, this review is motivated by the recent impact observed in both the literature and online.

Therefore, the review is focused on studies related to signals acquired from a human body in controlled conditions but exploring non-invasive, easily repeated measurements. For this purpose, EEG, heart rate, skin conductance, and eye-tracking signals were selected as the common denominator of research being conducted on ASMR. Moreover, investigating ASMR with EEG and fMRI (functional magnetic resonance imaging) is crucial for a comprehensive understanding of the phenomenon's neurological underpinnings. EEG provides real-time data on brain electrical activity, allowing researchers to observe immediate changes during ASMR experiences and identify involved brain regions and neural pathways. This helps to validate ASMR beyond subjective reports and differentiates it from similar states like meditation. fMRI complements EEG by offering high-resolution images of brain activity, pinpointing specific areas involved in ASMR with increased spatial accuracy. This combined approach reveals ASMR's therapeutic potential, particularly in stress reduction and mental health treatment. It does so by identifying brain wave patterns and activity changes associated with relaxation. Altogether, EEG and fMRI studies enhance our understanding of sensory processing and emotional regulation, paving the way for personalized and effective ASMR content. We will discuss these further in the following Section.

## SURVEY METHODOLOGY

Our study utilized keywords related to 'autonomous sensory meridian response' (ASMR) to search three prominent academic databases: PubMed, Scopus, and IEEE. In this research, we sought to identify relevant literature investigating ASMR phenomena using EEG and physiological measurements. After removing duplicate entries, the search strategy yielded 63 relevant articles from PubMed, 166 from Scopus, and 14 from IEEE. To ensure the relevance of the identified articles, we meticulously reviewed their titles and abstracts. The methodologies we explore are readily available in our laboratory apart from fMRI, which we decided to include to examine whether there exists a relationship between EEG and fMRI activity mapping, which may facilitate ASMR investigation. Due to this, we selected articles that directly addressed ASMR and its associated methodologies. We specifically sought studies utilizing electrophysiological and physiological measures (such as eye tracking, heart rate monitoring, skin conductance, *etc.*).

### Eligibility criteria

Before conducting the comprehensive search, we selected the following inclusion criteria (IC):

- IC1: Autonomous sensory meridian responses were examined.
- IC2: Articles from journals and conferences, including case reports, were considered.
- IC3: Only English-language articles were reviewed.

### Detailed methodology

To develop an integrative overview of ASMR definitions and assessment tools, we evaluated the individual ASMR articles in depth. Results associated with ASMR-related outcomes in the other areas covered in this review (electrophysiological monitoring and physiological ASMR correlates) were classified according to the following patterns.

ASMR-related brain activities;

- ASMR-related physiological changes;
- ASMR-related cognitive outcomes;
- Specific ASMR-related electroencephalogram (EEG) outcomes;
- Specific ASMR-related electroencephalogram (fMRI) outcomes;
- Specific ASMR-related heart rate (HR) outcomes;
- Specific ASMR-related skin conductance (SC) outcomes;
- Specific ASMR-related eye tracking outcomes.

## ASMR AND NEUROIMAGING TECHNIQUES

### Functional magnetic resonance imaging (fMRI)

Functional MRI is able to provide insight into ASMR's unique brain signatures. It compares them to other phenomena, such as relaxation or negative emotions. Therefore, the study explores differences in ASMR sensitivity among individuals and investigates how different triggers elicit different neural responses. Despite limitations such as cost and participant constraints, functional magnetic resonance imaging (fMRI) remains a valuable tool for advancing our understanding of ASMR. Our systematic search identified seven relevant studies in the context of fMRI research on ASMR. These studies employed two distinct methodological approaches. One approach aimed to identify connections between specific, isolated brain areas and ASMR stimulation, focusing on how individual brain regions respond to ASMR triggers (*Lohaus et al., 2023*). The other approach investigated the functional connectivity between multiple brain areas, examining how different regions of the brain interact and work together during ASMR experiences. This comprehensive exploration highlights the complexity of brain activity associated with ASMR, involving both isolated brain regions and broader neural networks.

In a recent study, *Sakurai et al. (2023)* demonstrates for the first time how direct audiovisual and auditory stimulation affect brain function using functional magnetic resonance imaging (fMRI). Furthermore, the study clarifies the effects of ASMR, which is particularly appealing to young people. An experiment was conducted on 30 healthy subjects, 15 males and 15 females aged 19 and over, who had viewed ASMR videos but

had not experienced tingling. As a precaution against habituation, the subjects were prohibited from watching ASMR for one week prior to the start of the experiment. In this study, auditory and audiovisual stimulation engaged different brain areas, but the mood questionnaire did not reveal any differences between them. Auditory stimulation activated the insular cortex, whereas audiovisual stimulation activated the middle frontal gyrus and nucleus accumbens. Auditory and audiovisual stimulation activate different brain regions, suggesting different mental health effects.

In another study that compared music to ASMR (*Sakurai et al., 2021*), a significant difference was found in participants experiencing a tingling sensation. In terms of brain function, both classical music and ASMR activated common areas, but ASMR activated additional regions, with the medial prefrontal cortex (mPFC) being the primary area of activation during ASMR. According to another study, participants underwent resting-state functional magnetic resonance imaging for eight minutes and were then required to complete the ASMR Checklist after the fMRI session. A cluster of voxels located in the posterior cingulate gyrus and precuneus was negatively correlated with the functional connectivity of the salience network (SN). The functional connectivity of the VIS was negatively correlated with the activity of voxels in the anterior cingulate cortex, paracingulate gyrus, and medial prefrontal cortex in the medial frontal region (*Smith, Fredborg & Kornelsen, 2020*).

Through seed-based correlation analysis, *Lee, Kim & Tak (2020)* discovered that functional connections between the posterior cingulate cortex and the superior/middle temporal gyri, cuneus, and lingual gyrus were significantly increased during ASMR compared to the resting state. Moreover, when examining the pregenual anterior cingulate cortex seed region, they found that functional connectivity in the medial prefrontal cortex was higher during ASMR than in the resting state. These findings suggest that ASMR can be initiated and maintained by continuous interactions between regional activities mainly involved in mentalizing and self-referential processing. *Lochte et al. (2018)* indicate that individuals experiencing ASMR exhibited notable activation in brain regions linked to reward nucleus accumbens (NAcc) and dorsal anterior cingulate cortex (dACC), insula/inferior frontal gyrus (dACC and Insula/IFG) and medial prefrontal cortex (mPFC). The brain activation observed during ASMR shares similarities with patterns seen in musical frisson and affiliative behaviors (*Lochte et al., 2018*). Furthermore, *Smith, Fredborg & Kornelsen (2017)* compared fMRI scans of 11 individuals experiencing ASMR with 11 matched controls. The findings revealed that the default mode network (DMN) in those who experienced ASMR exhibited less functional connectivity between the frontal lobes and sensory/attentional areas compared to the control group (see Table 2).

The main conclusion drawn from the studies on ASMR and its neural correlates is that ASMR experiences involve complex interactions within the brain, implicating various regions and networks. While specific brain areas like the anterior cingulate gyrus and movement-related regions show associations with ASMR responses, the overall functional connectivity patterns differ between ASMR responders and non-responders. Furthermore, alterations in connectivity during ASMR stimulation indicate dynamic neural processes underlying these experiences. These findings highlight the need for further research to

**Table 2  Functional MRI analysis of ASMR-induced brain activation and regional correlations.** The table shows a brief analysis of investigations related to fMRI and AMSR. Specifically, it shows increased activity (positive and negative) within certain brain regions.

| Article title | Source | Activated brain regions/positive or negative correlations |
| --- | --- | --- |
| Brain function effects of autonomous sensory meridian response (ASMR) video viewing | Sakurai et al. (2023) | Activity increased in insular cortex, middle frontal gyrus , nucleus accumbens (NAcc) |
| An fMRI investigation of the neural correlates underlying the autonomous sensory meridian response (ASMR) | Lochte et al. (2018) | Activity increased in nucleus accumbens (NAcc), dorsal anterior cingulate cortex (dACC), nsula/inferior frontal gyrus (IFG) and medial prefrontal cortex (mPFC) |
| An examination of the default mode network in individuals with autonomous sensory meridian response (ASMR) | Smith, Katherine Fredborg & Kornelsen (2017) | less functional connectivity between the frontal lobes and sensory/attentional areas |
| Functional connectivity associated with five different categories of Autonomous Sensory Meridian Response (ASMR) triggers | Smith, Fredborg & Kornelsen (2020) | Negative correlation between salience network (SN) functional connectivity with posterior cingulate gyrus and precuneus |
| Effects of Autonomous Sensory Meridian Response on the Functional Connectivity as Measured by Functional Magnetic Resonance Imaging | Lee, Kim & Tak (2020) | Higher functional activity between the posterior cingulate cortex and the superior/middle temporal gyri, cuneus, and lingual gyrus |
| Induction of Relaxation by Autonomous Sensory Meridian Response | Sakurai et al. (2021) | Higher activation in medial prefrontal cortex (mPFC) |

elucidate the neural mechanisms of ASMR and its potential implications for understanding brain sensory processing and relaxation mechanisms.

## Electroencephalography

The EEG (electroencephalography) provides a scientific basis for understanding the neural processes associated with autonomous sensory meridian response (ASMR). Research in this area could provide insights into brain function, emotional responses, and therapeutic methods. Given that ASMR research suggests this phenomenon could help with anxiety-related issues, most studies focus on investigating brain-emitted waves and which regions are affected by ASMR, thus benefiting further analysis, like brain stimulation. In the latest study by Sakurai et al. (2023) used Daubechies wavelets to filter EEG signals into the delta, theta, alpha, and beta components. The energy of each segment was computed, and normalized power was then calculated and compared with reference values. An individual with a theta value within the reference range may suffer from insomnia. A re-evaluation of the individual's brain signal values was performed following the use of ASMR, demonstrating improvement. Hence, based on these results, ASMR may enhance sleep quality and reduce insomnia.

EEG features of ASMR were examined using power spectral density (PSD) and differential entropy (DE) after segmentation with a 0.5-second rectangular window without overlap, as detailed by Yu et al. (2023). Statistical methods were employed to analyze PSD, DE, and scores from the NEO-PI-R's five personality dimensions, as well as scores from the Self-Rating Depression Scale (SDS) and Self-Rating Anxiety Scale (SAS). The Lilliefors test was first conducted to assess data normality. Depending on the distribution, differences were evaluated using one-way analysis of variance (ANOVA) for normally distributed
data or Mann–Whitney and Kruskal-Wallis tests for non-normally distributed data. The findings revealed that ASMR significantly impacts brain activity, with video triggers being more effective than auditory triggers, such as sand-cutting sounds. A significant association was found between ASMR and neuroticism, particularly its sub-dimensions of anxiety, self-consciousness, and vulnerability, as well as scores on the self-rating depression scale. However, this connection did not extend to emotions such as happiness, sadness, or fear. These results highlight the complex interplay between ASMR and individual differences in personality and emotional response.

In a study by *Engelbregt et al. (2022)*, artifact rejection was performed using the automated selection tool in NeuroGuide. F7, F8, P3, P4, T5, and T6 channels were selected to analyze alpha, theta, and beta frequency ranges. According to the Modified Combinatorial Nomenclature (MCN), T5 and T6 were renamed P7 and P8 for parietal measurements. The alpha frequency bands at channel T6 were significantly affected by group (tingles/no tingles) and condition (ASMR/control). A significant reduction in alpha power was observed at channel T6. Participants experiencing tingles experienced a marginal but significant decrease in alpha power at channel P4 compared to those experiencing no tingles. *Swart et al. (2022)* mentioned EEGLAB and OpenMEEG for processing and analyzing EEG data, showing that a decrease in high-frequency oscillations and an increase in low-frequency oscillations may be observed, causing ASMR.

In the study by Inagaki et al. (2022), FFT was applied every second to the entire dataset using a Hanning filter. An average of the resulting power frequency spectrum was calculated. Further, a probability distribution was derived from the determined power spectrum. After separating data into alpha-, gamma-, and high beta-band responses, an average was calculated for each frequency band. The Wilcoxon ranked sum test was used to assess significance, with Benjamini–Hochberg adjustments applied (*Kwong, Holland & Cheung, 2002*). When cognitive tasks induced mental workload rather than resting, alpha-band activity was reduced, and gamma (high beta)-band activity was elevated. However, introducing ASMR sound restored alpha and gamma-band activity to levels similar to resting during mental workload.

In the comparative study, *Lee et al. (2022)* compared the efficiency of ASMR and Binaural Beats (BB) on stress. Fast Fourier Transform was used for the spectral analysis. At each electrode, absolute power values were computed for five frequency bands. The relative power values were calculated as a percentage of the absolute power. Throughout the preprocessing and analysis, the NeuroGuide software was used. There were more pronounced changes in absolute beta power and high beta power among the ASMR group compared to the BB group. While both ASMR and BB are effective in reducing stress levels, ASMR can potentially increase beta and high beta waves associated with cortical arousal, unlike BB. A mixed-factor analysis of variance (ANOVA) was utilized to analyze data, with between-subjects factors including the subject group and within-subjects factors encompassing condition (ASMR *versus* control stimuli), time, and electrode channel. This approach allowed for a comprehensive examination of the interaction between these variables. Separate ANOVAs were conducted to assess the impact of stimulus type (audio *vs.* video) and frequency band (alpha, sensorimotor rhythm, theta, and gamma). Results
indicated that the ASMR group showed increased sensorimotor rhythms, alpha activity, and gamma activity, suggesting distinct neural activation patterns associated with ASMR experiences. Further analysis by *Pedrini, Marotta & Guazzini (2021)* treated ASMR as an idiosyncratic experience, employing multitaper discrete prolate spheroidal sequences (DPSS) to determine the characteristic power spectral density across different event states: Baseline, Relaxed, WeakASMR, and StrongASMR. Spectral smoothing of 2 Hz was applied between 1 and 80 Hz to achieve high-frequency resolution. The mean power for each participant was calculated by averaging segments within each state and normalizing by total power across 1-80 Hz. EEG analysis revealed that individuals experiencing ASMR exhibited distinct patterns of brain activity in the DMN, which includes regions such as the hippocampus, medial prefrontal cortex, temporal lateral cortex, temporal-parietal cortex, and posterior medial cortex. This finding underscores the complex neural dynamics underlying ASMR experiences and their localization within key brain networks.

A comprehensive data analysis using NeuroGuide software was conducted on electroencephalographic (EEG) signals sampled at 256 Hz, as detailed in the study by *Seifzadeh et al. (2021)*. The analysis incorporated a Z score threshold of 2 to facilitate automatic artifact rejection, which meticulously excluded drowsiness, eye movement, and muscle artifacts with high sensitivity. This methodological rigor ensured the integrity and reliability of the resulting data. The findings indicated a significant reduction in alpha-band power following ASMR video stimulation, whereas theta-band power exhibited no noticeable changes. Additionally, distinct alterations were observed in specific brain regions, highlighting the localized effects of ASMR stimuli.

To elucidate the differential brain responses to ASMR and "Funny" videos within the same subjects, variations in relative power during video viewing were assessed using EEG topography. The results demonstrated that viewing Funny videos generally elicited an increase in delta wave amplitudes while concurrently reducing gamma wave amplitudes. These findings were particularly pronounced in the frontal region (AFz, F1, Fz, F2) and central region (FC1, FCz, FC2, Cz, C2, CP1, CPz, CP2), where significant differences in delta-wave activity were recorded. Furthermore, gamma-wave activity in the occipital region (P7, PO7, PO3, O4, POz, Pz, P2, P4, P6, P8) showed substantial variations ($p = 0.0078$), as reported by *Koo et al. (2021)*. This intricate mapping of brain activity underscores the nuanced and region-specific neural dynamics elicited by different types of video stimuli, offering valuable insights into the underlying neural mechanisms.

In another EEG investigation of *Fredborg et al. (2021)*, The findings showed that ASMR triggers, especially those of the auditory nature, caused higher levels of alpha wave activity in individuals who reported experiencing ASMR themselves, as opposed to those in the control group. Additionally, there were noticeable rises in gamma waves and sensorimotor rhythm, aligning with the typical experiences associated with ASMR, characterized by a blend of attentional and sensorimotor aspects. Frequency measures were acquired using a Hanning window with 256 samples (2 s) at a 128 Hz sampling rate for EEG signals, as outlined by *Paszkiel, Dobrakowski & Łysiak (2020)*. The analysis focused on the peak alpha frequency (PAF), identified as the frequency with the highest spectral amplitude within the 7-15 Hz band. Notable increases were observed in the right centrotemporal (C4, T4), left
frontal (F3), and occipital (O1) regions, particularly near the PAF where the "eyes open" curve intersected the "eyes closed" curve. These findings suggest specific regional brain activity modulation associated with different states of visual attention.

Research indicates that music known to 'trigger' ASMR (*Del Campo & Kehle, 2016*), *e.g.*, music containing soft singing, gentle whispering vocals, or binaural beats, characterized by a slow tempo with calm, soothing melody, as well as relaxation music can both lessen stress (*Kovacevich & Huron, 2019*). Typically, the changes in brain activity caused by ASMR, particularly within the alpha and gamma frequency bands, could be beneficial for cognitive functions and reducing stress. Overall, ASMR seems to produce a distinct pattern of brain activity that affects areas linked to emotional regulation and self-awareness (see Table 3). This suggests that further investigation is worth exploring its potential therapeutic benefits.

The EEG studies exploring the neural mechanisms underlying ASMR offer valuable insights into its potential therapeutic benefits and neural correlates. ASMR triggers significant alterations in brain activity, particularly evident in the alpha and gamma frequency bands, indicative of its influence on emotional regulation and self-awareness. Research findings suggest that ASMR could potentially serve as a tool for stress reduction and cognitive enhancement. Moreover, ASMR-induced changes in brain activity may have applications in alleviating insomnia, enhancing sleep quality, and reducing anxiety-related issues. These results underscore the importance of further investigations into the therapeutic potential of ASMR and its underlying neural mechanisms.

## ASMR and autonomic and peripheral physiological measures

In exploring the physiological underpinnings of ASMR experiences, it is essential to consider the extensive literature surrounding the insula's role in interpreting internal bodily signals and translating them into subjective emotional states. This dual process involves the posterior insula representing objective interoceptive inputs and the anterior insula representing them as emotional experiences. Moreover, research suggests lateralization within the anterior insula, where positive emotions tend to localize on the left, while negative emotions reside predominantly on the right. This lateralization extends to the autonomic nervous system, with the left insula linked to parasympathetic control and the right insula to sympathetic activation (*Craig, 2002*; *Zaki, Davis & Ochsner, 2012*; *Beissner et al., 2013*). Aside from influencing brain activity, ASMR has also been linked to physiological effects. As we delved into the studies, we found that most ASMR-based articles focusing on physiology used heart rate and skin conductance level to measure the extent to which they are affected according to anxiety levels. As a physiological metric, heart rate offers valuable information about how the autonomic nervous system reacts to ASMR, indicating fluctuations in both arousal and relaxation. In the same way, skin conductance is a measure of the skin's electrical conductivity, mainly influenced by sweat gland activity. It is often used as a physiological indicator of sympathetic nervous system activity (*Kim & Lee, 2023*). Pupil diameter is another physiological parameter used in some ASMR-related studies. Changes in pupil diameter, known as pupillometry, can provide insight into anxiety's autonomic and emotional aspects (*Graur & Siegle, 2013*). Nevertheless, the association

**Table 3  ASMR-induced EEG changes: brain bands and regions.** A re-evaluation of the individual's brain signal values is performed following the use of ASMR, and the obtained results demonstrate improvement. Based on these results, ASMR may enhance sleep quality and reduce insomnia in this way. The changes in brain signal frequency, as shown in recent articles, are summarized in the table.

| Article title | Source | Frequency changes | | Brain region |
|---|---|---|---|---|
| | | Increase | Decrease | |
| The effects of autonomous sensory meridian response (ASMR) on mood, attention, heart rate, skin conductance and EEG in healthy young adults | Engelbregt et al. (2022) | – | Alpha | Temporal Region (T6) |
| | | | | Parietal Region (P4) |
| An electroencephalographic examination of the autonomous sensory meridian response (ASMR) | Fredborg et al. (2021) | Alpha | – | Sensorimotor |
| | | Gamma | | |
| ASMR amplifies low frequency and reduces high frequency oscillations | Swart et al. (2022) | Alpha | Delta | Occipital and parietal |
| | | Low-beta | Theta | Frontoparietal |
| Comparison of autonomous sensory meridian response and binaural auditory beats effects on stress reduction: a pilot study | Lee et al. (2022) | Beta | – | Both hemispheres |
| | | High-beta | | |
| Asmr as idiosyncratic experience: Experimental evidence | Pedrini, Marotta & Guazzini (2021) | – | Gamma | Both hemispheres |
| | | | Low-beta | |
| Cortical activation changes associated with autonomous sensory meridian response (asmr) | Seifzadeh et al. (2021) | Alpha | Delta | Central Regions |
| | | Beta | | Frontal Region |
| | | Gamma | | Frontoparietal |
| An approach for assessing arousal characteristics of ASMR using electroencephalographic power | Koo et al. (2021) | Alpha | – | Sensorimotor |
| | | Gamma | | |

between other physiological markers like cortisol and heart rate variability (HRV) has not been studied, and others, like respiration rate, have only been looked at very rarely.

Research by *Seifzadeh et al. (2023)* demonstrated that ASMR impacts both heart rate (HR) and skin conductance level (SCL). A single session of ASMR video watching resulted in a significant decrease in heart rate compared to baseline levels, while skin conductance showed a slight but non-significant reduction. Similarly, a study by *Poerio, Mank & Hostler (2022)* evaluated the effects of ASMR on HR and SCL, finding that participants who experienced ASMR displayed a decrease in heart rate, suggesting a positive association between ASMR and relaxation. Additionally, these individuals exhibited a higher SCL after watching ASMR videos, indicating that ASMR can induce both relaxation and excitement. This dual effect suggests that ASMR is associated with increased arousal as well.

Analogously, *Engelbregt et al. (2022)* found that "low conscientious" individuals observed decreased heart rate and increased electrodermal activity (EDA) during ASMR viewing compared to control in 38 young adults (33 females and five males). A recent study (*Villena-Gonzalez et al., 2023*) measured both the heartbeat counting task (HCT) and the electrophysiological index of interoceptive information, known as the heartbeat evoked potential (HEP), during the HCT and an ASMR tingle reporting task (ASMR-TRT). In both tasks, the ASMR group exhibited a larger HEP amplitude. As reflected in HEP, the ASMR experience relies on an unconscious interoceptive mechanism. As with affective touch, which is mediated by C-tactile afferent fibers, this mechanism integrates exteroceptive social-affective stimuli to depict a body state characterized by positive affective feelings. Unlike the usual myelinated tactile afferents, C-tactile afferents project *via* the thalamus to the dorsal posterior insula rather than to the primary somatosensory cortex. A detailed discussion of the theoretical mechanisms underlying ASMR is provided by *Craig (2002)* and *Villena-Gonzalez (2023)*.

In another study, ASMR was shown to students after they rested for 30 min, followed by three minutes of watching the video (*Idayati, Sufani & Syahputra, 2021*). Before and after ASMR, heart rate, blood pressure, and respiratory rate were calculated. Wilcoxon test data analysis showed significant decreases in HR and blood pressure (BP) following viewing the ASMR video, while no significant differences in respiratory rate were observed. A study by *Carlaw et al. (2022)* investigated ASMR's effects on pre-operative anxiety in a double-blind trial with 50 participants. The ASMR group showed a significant reduction in anxiety and systolic blood pressure, while the control group improved in anxiety measures.

Utilizing finger photoplethysmography (PPG), *Tada, Ezaki & Kondo (2021)* examined ASMR responses. They found that when auditory and visual stimuli were presented together, ASMR intensity surpassed that of auditory stimuli alone, suggesting the involvement of sensory-based, bottom-up processing. Additionally, participants exhibited decreased pulse rates (PR) and increased PPG amplitudes during ASMR, indicating physiological indicators of relaxation and heightened blood flow. In the study by *Valtakari et al. (2019)*, they examined participants in two groups, namely experience groups and non-experience groups, using eye-tracking while showing ASMR and control videos. Noting any tingling sensations required pressing a key on the keyboard. Throughout both videos, pupil diameter was measured using an eye tracker. Data were analyzed generally, averaging pupil diameters over each video, and in detail, comparing pupil diameters during reports of tingling to those without. The diameter of the pupils increased statistically significantly after episodes of tingling sensations induced by ASMR, which were not significant at a general level. Pedrini et al. analyzed pupil diameter and brain activity (*Pedrini, Marotta & Guazzini, 2021*) to examine ASMR as an idiosyncratic experience. Rest, control, and ASMR videos were compared. This was the largest difference in pupil diameter between the ASMR and rest videos; and more importantly, the difference was the most significant. ASMR and control videos showed the smallest pupil diameter difference. Despite perceptions of tingles, an increased pupillary diameter was observed in response to the ASMR video. Table 4 provides a brief analysis of physiological changes.

**Table 4   ASMR-induced physiological changes.** Table 4 provides a brief analysis of physiological changes, *i.e.*, heart rate (HR), heartbeat0 evoked potential (HEP), blood pressure (BP), pules rates (PR), ûnger photoplethysmography (PPG), pupil diameter, and skin conductance (SC) that occurred after experiencing ASMR.

| Article title | Source | Physiological changes | |
| --- | --- | --- | --- |
| | | Increase | Decrease |
| The Physiological Effects of ASMR on Anxiety. Frontiers in Biomedical Technologies | *Seifzadeh et al. (2021)* | – | HR |
| Autonomous sensory meridian response (ASMR) is characterized by reliable changes in affect and physiology | *Poerio, Mank & Hostler (2022)* | SC | HR |
| Autonomous sensory meridian response is associated with a larger heartbeat-evoked potential amplitude without differences in interoceptive awareness | *Villena-Gonzalez et al. (2023)* | HEP | – |
| Effect of watching autonomous sensory meridian response (AMR) video to heart rate, blood pressure and respiratory rate in students of Architectural Engineering, Universitas Syiah Kuala, Banda Aceh, Indonesia | *Idayati, Sufani & Syahputra (2021)* | – | HR |
| | | | BP |
| The Autonomous Sensory Meridian Response Activates the Parasympathetic Nervous System | *Tada, Ezaki & Kondo (2021)* | PPG | PR |
| An eye-tracking approach to autonomous sensory meridian response (ASMR): The physiology and nature of tingles in relation to the pupil | *Valtakari et al. (2019)* | Pupil Diameter | – |
| Asmr as idiosyncratic experience: Experimental evidence | *Pedrini, Marotta & Guazzini (2021)* | Pupil Diameter | – |
| Does Triggering an Autonomous Sensory Meridian Response Reduce Pre-Operative Anxiety? A Randomized Placebo Controlled Trial | *Carlaw et al. (2022)* | – | BP |

In conclusion, the investigation of physiological responses to ASMR provides valuable insights into its effects on the autonomic nervous system and emotional arousal. Studies examining heart rate, skin conductance level, and pupil diameter reveal a complex interplay between relaxation and arousal induced by ASMR stimuli. Research consistently demonstrates that ASMR leads to decreases in heart rate, indicative of relaxation, while potentially increasing skin conductance level, suggesting concurrent excitement. Moreover, changes in pupil diameter following ASMR episodes further highlight its influence on emotional and autonomic processes. The findings underscore the therapeutic potential of ASMR in promoting relaxation and emotional well-being. At the same time, it is indicated that further research into these physiological markers is crucial to understand the underlying mechanisms better and optimize the ASMR's therapeutic benefits. Overall, exploring physiological responses to ASMR contributes to a deeper understanding of its physiological effects and potential applications in stress reduction and emotional regulation.

## CONCLUSIONS

According to our review of studies on ASMR, specific physiological and neural responses were consistently observed. As a result of ASMR, subjects experience a decrease in heart rate, which indicates a reduction in autonomic arousal. Moreover, skin conductance and pupil diameter growth are observed, reflecting increased emotional arousal and involvement. Multiple EEG analyses indicate a significant increase in alpha band activity, particularly within the sensorimotor cortex. An elevated level of alpha activity in this region of the brain indicates a state of rest or idling, as well as a decreased level of motor activity. It is consistent with periods of calm wakefulness and no motor demands.

Collectively, these findings point to ASMR's dual nature, combining physiological relaxation and emotional arousal. It is apparent that the consistent increase in alpha activity within the sensorimotor cortex contributes to the calming and tingling sensations experienced by individuals experiencing ASMR. Therefore, ASMR appears to induce a distinctive combination of sensory-emotional engagement and relaxation, accompanied by identifiable physiological and neural characteristics. In addition, based on the results of the fMRI investigation articles, it appears that ASMR involves increased activity in specific brain regions such as the insular cortex, middle frontal gyrus, NACC, Insula/IFG, and medial prefrontal cortex (mPFC), suggesting a broader involvement of reward, emotional processing, social cognition, and self-referential processing regions suggesting a multifaceted neural response to ASMR stimuli. Overall, ASMR elicits a unique combination of sensory-emotional engagement and relaxation, characterized by identifiable physiological and neural signatures.

However, continued exploration of the physiological effects of ASMR, alongside efforts to delineate its neural mechanisms, will be instrumental in unlocking its therapeutic potential as they may complement clinical interventions. Ethical and social implications associated with ASMR study and practice should also be considered. This includes issues such as consent in ASMR content creation but also potential risks of over-reliance on ASMR for coping with anxiety and stress.

### Funding
The authors received no funding for this work.

### Competing Interests
The authors declare there are no competing interests.

### Author Contributions
- Sahar Seifzadeh conceived and designed the experiments, performed the experiments, analyzed the data, prepared figures and/or tables, authored or reviewed drafts of the article, and approved the final draft.
- Bozena Kostek conceived and designed the experiments, performed the experiments, authored or reviewed drafts of the article, and approved the final draft.

## Data Availability

This is a literature review.

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
