# Peer review of "Exploring the technological dimension of Autonomous sensory meridian response-induced physiological responses"

_PeerJ, doi:10.7717/peerj.17754_

## Round 0.1 · original submission · Major Revisions

Your manuscript has now been seen by 3 reviewers. You will see from their comments below that while they find your work of interest, some major points are raised. We are interested in the possibility of publishing your study, but would like to consider your response to these concerns in the form of a revised manuscript before we make a final decision on publication. We therefore invite you to revise and resubmit your manuscript, taking into account the points raised. Please highlight all changes in the manuscript.

Reviewer 1 ·

Basic reporting

Most sentences are clear, I suggest some changes in the additional comments.

References and background information needs improvement, see my additional comments.

Figures and tables are very well done, with one major change suggested in my additional comments.

Yes, paper is suitable for journal.

Yes, paper adds to the current body of knowledge, especially their figures.

No, the introduction is not satisfactory and does need improvement. See my additional comments.

Experimental design

Yes, paper fits the journal.

Yes, Mostly proper investigation was performed, see my comments for suggested improvements.

Methods need some Improvements, see my comments.

Many statements are missing citations, see my comments.

Introduction is not well organized, it is too lengthy, and has some awkward opinions. See my comments for suggested improvements.

Validity of the findings

Paper is beneficial to the field.

The conclusions are well stated.

The argument/introduction is not well stated, see my comments for suggestions.

Additional comments

Aim of paper: Very good and helpful to the field of ASMR research.

Figures and Tables: Extremely good and helpful to the field of ASMR research (with one change I suggest below). Accolades to the authors for these tables, well done.

Text and content of the paper: Mostly good but need a lot of minor improvements.

Below are my suggested changes to each section to help improve the clarity, flow, and usefulness of this paper.

Title, Abstract, Introduction:

“Autonomous sensory meridian response (ASMR)”, check with editor or style guide to see if all words should be capitalized, none, or some.

Abstract:

“The paper uses technology-based methodologies” paper is a review article so the paper doesn’t “use” these methods but it compiles, investigates, reports, (or pick other appropriate term).

“…using EEG and physiological measurements.” and “…of EEG and physiological measures” Please explain to the readers why EEG is excluded as a physiological measurement or change this statement. According to others, EEG is a physiological measurement so some readers will be confused, see this paper: https://www.ncbi.nlm.nih.gov/pmc/articles/PMC6696017/

“These studies were selected for their contribution to understanding ASMR's effects on electrophysiological and physiological responses.” Confusing bc it sounds like ASMR is affecting these methods. Reword to clarify that the methods are being used to understand ASMR.

Introduction

Missing: an explanation that ASMR videos are just simulating real-world ASMR stimuli/scenarios. Authors need to explain that ASMR happens all the time in real life through sound, visuals, and touch. ASMR videos are just recreating those real-world moments.

Overall: Introduction should be condensed a bit more to remove the casual, non-science-like writing. Should be more focused on providing key background details and facts, and less focused on opinions.

Line 58: “purported to calming” is missing a word?
Lines 57-59: Statement only lists auditory and visual cues, missing tactile cues.
Line 59-60: Awkward statement about motivation, reword or remove.
Line 60: “As of December 2023, ASMR” specify the exact source and meaning of this date.
Lines 60-61: needs a reference, reword for accuracy and/or add a reference.
Lines 62-63: needs a reference
Line 65: needs a specific year and a reference
66-68: needs a reference
74-75: strange opinion, could offend some readers who disagree, reword.
79-84: wordy and perhaps unnecessary, suggestion to remove or condense.
85-90: these 3 sentences don’t relate directly to each other, please reorganize introduction so each paragraph has a thematic focus.
85-136: paragraph is way too long, should be broken up into smaller thematic paragraphs
93-94: needs a reference and statement is not true as written, those percents don’t apply to all publications and experiences with ASMR, those percents only apply to the paper which is not referenced.
96: change “this study” to “a particular study” or something similar.
117: missing tactile stimuli
119: consult with editor to see if “(which will be discussed later)” is appropriate.
121: partial acronym for fMRI is typed out after fMRI has already been used.
137-139: add the references to each statement
140: change “are not measured” to “have not yet been measured”
140: suggestion to remove statement, “Below, we will review…”
137-162: too long, repetitive of earlier statements, and mostly unnecessary. Suggestion to condense these 3 paragraphs down to one, focused paragraph.

Survey Methodology

166-171: this description of the search terms seems more vague (and misleading?) compared to the method description in the abstract. Please rewrite this statement to be more like the statement in the abstract.
176: “Sensory meridian responses” typo, please fix to clarify it was ASMR
177: wrong verb, change “are” to “were”
181-195: seems repetitive to prior information and mostly unnecessary, please remove or condense

Remainder of paper:

230-231: list out the other triggers to provide clarity to reader
290-291: same reference put twice?
296: I suggest putting ASMR-triggering music in quotes bc it is not a confirmed thing. The authors of that paper don’t state what the “ASMR-triggering music” was, how it was determined to trigger ASMR, and don’t cite any papers to show that music is a confirmed trigger for ASMR.
311-314: put references.
314: put reference
325: typo, “ASMR watching” should be ASMR video watching

Table 1: paper discusses fMRI studies but table is missing information about fMRI and brain scans?

Table 2 and Table 3: very helpful, I encourage the inclusion of a similar table about fMRI

Table 2 & Table 3: Column with “Paper Title” needs to show consistent information about papers that also allows reader to easily find the paper. I suggest changing column title to “Source” or “Reference”, then putting first author and year so that readers can easily find the reference in the list of references.
Table 3: I think the paper mentioned more than one SC result, but the table only shows one SC result. Please review and include more SC results into table if appropriate.

References:

Is this a missing reference that could be included?
An electroencephalographic examination of the autonomous sensory meridian response (ASMR).
Fredborg BK, Champagne-Jorgensen K, Desroches AS, Smith SD.
Conscious Cogn. 2021 Jan;87:103053. doi: 10.1016/j.concog.2020.103053. Epub 2020 Nov 21.
PMID: 33232904

Here is a Relevant Reference that may not have been in your search databases, but perhaps could still be mentioned in your paper:

(ASMR and Blood Pressure) Does Triggering an Autonomous Sensory Meridian Response Reduce Pre-Operative
Anxiety? A Randomized Placebo Controlled Trial
Kirsten R Carlaw1,2*, David Tok Fu Ng1,2, Dukyeon Kim1,3,4 and Stephanie Phillips1,
ANESTHESIA & CLINICAL RESEARCH |ISSN 2733-2500
https://www.sciencerepository.org/articles/does-triggering-an-autonomous_ACR-2022-1-102.pdf

A note about the exclusion/non-focus of fMRI studies:

It appears that the authors have decided to include EEG but not fMRI studies in their method search. This is OK, but they should address and explain that decision in the introduction. This is because they mention some fMRI studies in their paper, which can confuse some readers. It would be wonderful if they expanded this review paper to include fMRi studies and made a fMRI table similar to their EEG table. But I do understand that that may not be their intent or objective for this paper.

If the authors decide to expand the paper to include a focus on fMRI studies, then here are some additional references:

Induction of Relaxation by Autonomous Sensory Meridian Response
Noriko Sakurai1 Ken Ohno1 Satoshi Kasai1 Kazuaki Nagasaka2 Hideaki Onishi2 Naoki Kodama1*
JOURNAL=Frontiers in Behavioral Neuroscience
VOLUME=15
YEAR=2021
URL=https://www.frontiersin.org/articles/10.3389/fnbeh.2021.761621
DOI=10.3389/fnbeh.2021.761621
ISSN=1662-5153
https://www.frontiersin.org/articles/10.3389/fnbeh.2021.761621/full

Atypical Functional Connectivity Associated with Autonomous Sensory Meridian Response: An Examination of Five Resting-State Networks.
Smith SD, Fredborg BK, Kornelsen J.
Brain Connect. 2019 Jul;9(6):508-518. doi: 10.1089/brain.2018.0618. Epub 2019 May 7.
PMID: 30931592

A functional magnetic resonance imaging investigation of the autonomous sensory meridian response.
Smith SD, Fredborg BK, Kornelsen J.
PeerJ. 2019 Jun 21;7:e7122. doi: 10.7717/peerj.7122. eCollection 2019.
PMID: 31275748

Tingles down the spinal cord: A spinal functional magnetic resonance imaging investigation of the autonomous sensory meridian response.
Smith SD, Kolesar TA, Fredborg BK, Kornelsen J.
Perception. 2022 Jul;51(7):514-517. doi: 10.1177/03010066221098964. Epub 2022 May 16.
PMID: 35578557

Structural differences in the cortex of individuals who experience the autonomous sensory meridian response.
Kornelsen J, Fredborg BK, Smith SD.
Brain Behav. 2023 Feb;13(2):e2894. doi: 10.1002/brb3.2894. Epub 2023 Jan 24.
PMID: 36692975

Reviewer 2 ·

Basic reporting

Review of the article “Exploring the technological dimension of Autonomous sensory meridian response-induced physiological responses (#97387)”
The manuscript provides a literature review focusing on the electrophysiological and physiological effects of the Autonomous Sensory Meridian Response (ASMR) as studied through EEG and various physiological measures. This is an intriguing area of research, given the growing interest in ASMR's potential therapeutic applications.
The text is clear and includes up-to-date references. I also believe that the topic fits well within the scope of the journal. However, concerning Basic Reporting, I have a major concern. In 2023, a systematic review guided by the PRISMA methodology on the topic of ASMR was published (Lohaus, Schreckenberg, Bellingrath, & Thoma, 2023). This mentioned review, which is not cited in the current manuscript, also addresses the physiological effects of ASMR. Therefore, I would like the authors to address how the current review differs from and stands out against the recent review mentioned, particularly highlighting its necessity for the scientific community. This ties in with clarifying the motivation for approaching this topic from the perspective of "advanced technologies," which is not very clear throughout the text.

Lohaus, T., Schreckenberg, S. C., Bellingrath, S., & Thoma, P. (2023). Autonomous sensory meridian response (ASMR): A PRISMA-guided systematic review. Psychology of Consciousness: Theory, Research, and Practice. https://doi.org/10.1037/CNS0000368

Experimental design

Regarding the study design, I am not an expert in reviews, but the protocol seems consistent with what is shown in similar reviews. However, typically reviews use existing methodologies, like PRISMA, for example. While the review appears rigorous in terms of the articles covered, the omission of the 2023 systematic review seems negligent. It is uncommon for reviews to be published so closely together, and there must be a very significant justification for a review to be published the year following another review on the same topic

Validity of the findings

I must confess that I am particularly interested in this compilation of information regarding physiological measures associated with ASMR. However, despite my interest, reading the article was challenging because it reads more like a collection of information rather than a well-constructed narrative with a clear motivation. I believe the authors could significantly improve the structure to make the reading more fluid and clearly convey the purpose of the review. In this regard, the introduction does not clearly justify the motivation for the review, the conclusions are weak, and the body of the article is a compilation of information without systematically addressing the results of the analyzed papers.

Additional comments

Line 57: It is a sensory-emotional phenomenon, rather than merely a perceptual one.
Line 85: ASMR instead of ARMR.
Line 90: The parentheses in the "del Campo" citation should be consistent with the rest of the citations.
Line 110: Interoceptive instead of interception.
Line 168: A period is missing between "study" and "After."
Line 342: The theoretical mechanism mentioned is explained in greater detail in
Villena-Gonzalez, M. (2023). Caresses, whispers and affective faces: A theoretical framework for a multimodal interoceptive mechanism underlying ASMR and affective touch. BioEssays, 45(12), 2300095. https://doi.org/10.1002/bies.202300095

Reviewer 3 ·

Basic reporting

I think that the aspiration of the paper to review and collate the physiological changes observed in individuals with ASMR is a worthy one and would make a worthwhile contribution to the literature.

I found the section on ASMR and Electrophysiological Monitoring (starting line 211) to be dense and confusing. I suggest restructuring this section to review the various studies in chronological order and sticking to one per paragraph.

Experimental design

Line 176: the authors state that inclusion criterion number 1 is that "Sensory meridian responses were examined." The phenomenon was named Autonomous Sensory Meridian Response by a member of an online forum in 2010. However, sensory meridians are not generally regarded as being real. So, what do the authors mean that an examination of sensory meridians was an inclusion criterion? I suggest clarifying what they did here to bring it in line with modern scientific thinking.

Validity of the findings

Line 333 the authors refer to a study of "low conscious" individuals. Please could the authors explain what this study meant by this?

Additional comments

At line 367 the authors start the Conclusions. I would like to see a bit more in the way of substantive discussion of the findings in this section. At line 340 the authors briefly mention affective touch. This sensory modality is mediated by C-tactile afferent fibres, rather than the usual myelinated tactile afferents, and the input from them project via the thalamus, not to the primary somatosensory cortex, but to the dorsal posterior insula.

Indeed, there is a large literature on the role of the insula in representing objective interoceptive inputs on the physiological state of the body posteriorly and then re-representing them as subjective emotional states in the anterior insula. Moreover, there is evidence that this representation anteriorly is lateralized with positive, affiliative, bonding type emotions on the left and negative, challenging emotions on the right. And that this lateralization includes the autonomic nervous system with the left involved in parasympathetic control and the right in sympathetic. This all is directly relevant to why ASMR individuals may be manifesting these physiological changes and thus worthwhile for the authors to consider.

If the authors look at these papers then I think it will help aid their thinking on how to frame this discussion:

Craig AD. How do you feel--now? The anterior insula and human awareness. Nat Rev Neurosci. 2009 Jan;10(1):59-70. doi: 10.1038/nrn2555. PMID: 19096369.

McGeoch PD, Rouw R. How everyday sounds can trigger strong emotions: ASMR, misophonia and the feeling of wellbeing. Bioessays. 2020 Dec;42(12):e2000099. doi: 10.1002/bies.202000099. Epub 2020 Nov 10. PMID: 33174254.

Craig AD. Forebrain emotional asymmetry: a neuroanatomical basis? Trends Cogn Sci. 2005 Dec;9(12):566-71. doi: 10.1016/j.tics.2005.10.005. Epub 2005 Nov 4. PMID: 16275155.

---

## Round 0.2 · accepted · Accept

Thank you for the revised manuscript and response letter. I am pleased to inform you that your manuscript "Exploring the technological dimension of Autonomous sensory meridian response-induced physiological responses" has been accepted for publication in PeerJ. However, before publication, please take into account the important corrections indicated by Reviewer 2.

Reviewer 1 ·

Basic reporting

Authors addressed my initial concerns and made appropriate improvements.

Experimental design

Authors addressed my initial concerns and made appropriate improvements.

Validity of the findings

Authors addressed my initial concerns and made appropriate improvements.

Additional comments

Authors addressed my initial concerns and made appropriate improvements.

Reviewer 2 ·

Basic reporting

On lines 397 and 399, the word "interceptive" should be corrected to "interoception" and "interoceptive," respectively. Please remember to include the 'o'.
Although you included the suggested citations in the text, they were not added to the bibliography. Please review and update the reference list accordingly.
Overall, the authors have addressed my previous concerns adequately. Apart from the issues mentioned above, I have no further comments.

Experimental design

no comment

Validity of the findings

no comment

Additional comments

no comment

Reviewer 3 ·

Basic reporting

Clear and professional.

Experimental design

Appropriate

Validity of the findings

Meaningful contribution to the literature.

Additional comments

The authors have positively engaged with and successfully addressed my comments.